# Evaluation of the Performance of Large Language Models in the Management of Axial Spondyloarthropathy: Analysis of EULAR 2022 Recommendations

**DOI:** 10.3390/diagnostics15121455

**Published:** 2025-06-07

**Authors:** Ahmet Usen, Ozlem Kuculmez

**Affiliations:** 1Department of Physical Medicine and Rehabilitation, Medipol University, Istanbul 34810, Turkey; ahmetusen1@hotmail.com; 2Department of Physical Medicine and Rehabilitation, Baskent University Alanya Hospital, Antalya 07400, Turkey

**Keywords:** artificial intelligence, guideline, rheumatic disease, spondylarthritis

## Abstract

**Introduction**: Guidelines have great importance in revealing complex and chronic conditions such as axial spondyloarthropathy. The aim of this study is to compare the answers given by various large language models to open-ended questions created from ASAS–EULAR 2022 guidance. **Materials and Methods**: This was a cross-sectional and comparative study. A total of 15 recommendations in the ASAS–EULAR 2022 guideline were derived directly from their content into open-ended questions in a clinical context. Each question was asked to the ChatGPT-3.5, GPT-4o, and Gemini 2.0 Flash models, and the answers were evaluated with a seven-point Likert system in terms of usability, reliability, Flesch–Kincaid Reading Ease (FKRE) and Flesch–Kincaid Grade Level (FKGL) metrics for readability, Universal Sentence Encoder (USE) and ROUGE-L for semantic and surface-level similarity. The results of different large language models were statistically compared, and *p* < 0.05 was revealed as statistically significant. **Results**: Better FKRE and FKGL scores were obtained in the Google Gemini 2.0 program (*p* < 0.05). Reliability and usefulness scores were significantly higher for ChatGPT-4o and Gemini 2.0 (*p* < 0.05). ChatGPT-4o yielded significantly higher semantic similarity scores compared to ChatGPT-3.5 (*p* < 0.05). There was a negative correlation between FKRE and FKGL scores and a positive correlation between reliability and usefulness scores (*p* < 0.05). **Conclusions**: It was determined that ChatGPT-4o and Gemini 2.0 programs gave more reliable, useful, and readable answers to open-ended questions derived from the ASAS–EULAR 2022 guidelines. These programs may potentially assist in supporting treatment decision-making in rheumatology in the future.

## 1. Introduction

Axial spondyloarthropathy (axial SpA) is a chronic rheumatic disease characterized by inflammation of the spine, sacroiliac joint, and peripheral joints. It is generally detected in young adults, more frequently in the male gender [1]. The most frequent initial symptoms are lower back pain, morning stiffness, and limitation in spinal movements. The diagnosis of axial spondyloarthritis is often challenging and may be delayed by several years, with reported delays of up to ten years in some cases [2]. Accurate diagnosis and effective management of axial SpA can improve patients’ quality of life and prevent future damage. Therefore, effective management of the disease has great importance [3].

Guidelines have great importance in managing complex and chronic conditions such as axial SpA. Guidelines that were developed by international rheumatology societies, such as the European League Against Rheumatism (EULAR) and Assessment of SpondyloArthritis international Society (ASAS), play a critical role in making decisions by both patients and healthcare professionals. The EULAR guidelines recommend appropriate treatment strategies for different disease activity levels and manifestations. This helps take control of disease activity and improve patients’ quality of life. EULAR provides recommendations for controlling disease activity, regular follow-ups, and pharmacological and non-pharmacological alternatives for disease management. This provides a standard for monitoring progress and preventing side effects [4,5,6].

Chat Generative Pre-trained Transformer (ChatGPT) and Gemini are the most common large language models (LLMs) that utilize human feedback and reinforcement learning. They have been preferred by millions of people because of their fast answers and easy access and become more popular day by day. There are previous studies evaluating the accuracy and reliability of large language models’ answers to frequently asked questions about public health [7,8]. Artificial intelligence has recently started to be used in rheumatology. Individuals, including both medical staff and patients, have started using it to explore potential conditions by inputting their symptoms. Laboratory data have also been added to request diagnostic suggestions. Advancing further, users have begun uploading radiological scans, prompting the systems to provide a ranked list of likely diagnoses [9,10,11,12]. In this study, we wanted to evaluate how adequately and accurately different large language models (such as ChatGPT-3.5, ChatGPT-4o Plus, and Gemini) provide answers to questions about important issues in the management of axial spondyloarthropathy in light of valid and up-to-date guidance. The aim of this study is to compare the responses of artificial-intelligence-based large language models (ChatGPT-3.5, ChatGPT-4o, and Gemini) in terms of usability, reliability, readability, and semantic and surface-level similarity through open-ended questions derived from the axial spondyloarthritis management recommendations published by the EULAR in 2022. In addition, this study aimed to qualitatively examine the extent to which these models offer accurate guidance to patients, alongside the comprehensiveness and accuracy of the information they deliver.

## 2. Materials and Methods

This study was designed as a cross-sectional and comparative study. Ethical approval was obtained from the Medipol University Ethics Committee (approval number: E-10840098-202.3.02-2713, date: 17 April 2025) and this study was conducted in accordance with the Declaration of Helsinki.

### 2.1. Open-Ended Question Development Process

A total of 15 recommendations from the ASAS–EULAR 2022 guidelines were translated into open-ended questions in a clinical context, derived directly from their content, like in Liu et al.’s study [13]. This process was conducted by two physical medicine and rehabilitation specialists with clinical experience in axial spondyloarthritis. Care was taken to preserve the original clinical intent of each recommendation during the formulation of the questions. Each item was designed to reflect the core message of its corresponding recommendation. The anticipated responses are based on the original content of the guideline.

The questions address a range of clinical themes, including treatment individualization, disease monitoring, patient education, pharmacological and non-pharmacological interventions, treatment response scenarios, and surgical options. Accordingly, the questions were grouped into four thematic categories, as in the study by Sciberras et al. [14]: (1) general management and monitoring strategies, (2) pharmacological treatment, (3) treatment response and adaptation strategies, and (4) surgery and accompanying conditions. This classification is presented in Table 1, and open-ended questions created from the ASAS–EULAR 2022 guidelines are given in Table 2.

### 2.2. Large Language Models Used and Response Collection Process

In the study, large language models, ChatGPT-3.5 (Google LLC, Alphabet Inc., America, California), ChatGPT-4o (Google LLC, Alphabet Inc., America, California), and Gemini 2.0 Flash (Google LLC, Alphabet Inc., California), were used. These are large language models trained on extensive textual data, allowing them to generate responses that are contextually relevant and coherent, resembling those of a skilled human reader [15]. ChatGPT-3.5 and ChatGPT-4o (plus) are recent popular LLMs created by OpenAI, like Gemini was formed by Google [16]. Each open-ended question was submitted to the models in separate and freshly initiated sessions, with no prior context. The standardized prompt used across all models was as follows: “Please answer without considering the previous conversation.” The generated responses were recorded as text and anonymized for subsequent evaluation.

### 2.3. Evaluation Process of Responses

The answer given by each model to each question was independently evaluated by two experienced physiatrists (AU, OK), according to the reliability and usefulness criteria using the 7-point Likert scale developed by Uz et al. [17]. In cases where the evaluators disagreed, a consensus was first sought. If a consensus could not be reached, a third referee, who was also an experienced physiatrist specialized in the field (AA), evaluated the response. The 7-point Likert scale for the evaluation of reliability and usefulness is presented in Table 3. It has been known that LLMs like ChatGPT and Gemini are updated frequently; thus, reproducibility may be affected over time. To mitigate this issue, each question was presented to the models only once. The responses were printed in duplicate and distributed to the specialists, who evaluated the models independently, without knowledge of which response corresponded to which model.

In this study, the semantic similarity between model-generated responses and the ASAS–EULAR 2022 recommendations was evaluated using the Universal Sentence Encoder (USE). USE encodes texts into 512-dimensional vectors; this dimensionality reflects an empirically optimized balance between semantic representation and computational efficiency [18]. The transformer-based version was applied to both model outputs and reference texts, followed by cosine similarity calculation. Scores range from 0 to 1, with values closer to 1 indicating higher semantic similarity. All computations were performed in Python (v3.10) using the tensorflow_hub and scikit-learn libraries [19]. The surface-level similarity between model responses and the recommendations was assessed using the ROUGE-L metric, which measures the longest common subsequence (LCS) to capture word order and lexical overlap. ROUGE-L F1 scores were computed via the rouge-score package in Python; higher scores indicate greater lexical alignment between the texts [20,21].

Additionally, to evaluate the readability of the model-generated responses, we used the Flesch–Kincaid Reading Ease (FKRE) and Flesch–Kincaid Grade Level (FKGL) metrics. FKRE reflects ease of reading, with higher values indicating more readable text, while FKGL estimates the U.S. school grade level required to understand the text [22]. The FKRE score is calculated based on the average sentence length and the number of syllables per word, where higher scores reflect easier-to-read text. The FKGL score represents the U.S. school grade level required to comprehend the content [23]. These metrics were calculated using the textstat package in Python. (FKRE = 206.835 − 1.015 × (total words/total sentences) − 84.6 × (total syllables/total words) FKGL = 0.39 × (total words/total sentences) + 11.8 × (total syllables/total words) − 15.59 ROUGE-L F1 = [(1 + β^2^) × P × R]/[R + β^2^ × P], where P = LCS/m, R = LCS/n, and β = P/R)

### 2.4. Statistical Analysis

All statistical analyses were performed using IBM SPSS Statistics for Windows, Version 21.0 (IBM Corp., Armonk, NY, USA). The normality of continuous variables was assessed using the Shapiro–Wilk test. For variables exhibiting a normal distribution, the results were expressed as the mean and standard deviation; in contrast, for those not conforming to a normal distribution, the data were summarized using the median and range (minimum–maximum). Pairwise comparisons between the three large language models (ChatGPT-3.5, ChatGPT-4o, and Gemini 2.0) were performed using the Paired Samples’ t-test or the Wilcoxon Signed-Rank Test, depending on the normality of the data. Associations between numerical variables were examined using Spearman’s rank-order correlation. A *p*-value of <0.05 was considered statistically significant, which is a widely accepted threshold in biomedical research to control for Type I error while maintaining statistical sensitivity.

## 3. Results

Descriptive statistics for each large language model are presented in Table 4. In terms of FKGL and FKRE values, the Gemini 2.0 model produced text that was statistically more readable compared to the other models. For FKGL, both ChatGPT-4o [17.1 (15–20.3)] and Gemini 2.0 [15.7 (13.6–17.9)] had significantly lower scores compared to ChatGPT-3.5 [17.8 (14.6–19)]. (*p* < 0.001 and *p* = 0.003, respectively). Regarding FKRE, ChatGPT-4o [4.17 (–2.94–16.73)] and Gemini 2.0 [15.31 (1.13–27.01)] produced significantly higher scores than ChatGPT-3.5 [6.74 (–0.1–20.48)] (both *p* < 0.001). The reliability scores were significantly higher for ChatGPT-4o [6 (5–7)] and Gemini 2.0 [6 (4–7)] compared to ChatGPT-3.5 [4 (3–6)] (*p* < 0.001 and *p* = 0.001, respectively). Similarly, the usefulness scores were significantly higher for ChatGPT-4o [7 (5–7)] and Gemini 2.0 [7 (4–7)] than for ChatGPT-3.5 [5 (4–7)] (*p* = 0.002 and *p* < 0.001, respectively). Additionally, ChatGPT-4o [68.89 (53.64–79.57)] yielded significantly higher semantic similarity scores compared to ChatGPT-3.5 [66.19 (50.83–76.61)]. (*p* = 0.038). In terms of ROUGE-L F1 scores, Gemini 2.0 [13.3 (8.3–17.1)] yielded significantly lower values compared to ChatGPT-3.5 [13.9 (10.7–18.6)]. (*p* = 0.025), while no significant difference was observed between ChatGPT-4o [14.4 (10.8–21.2)] and Gemini 2.0 (*p* = 0.074). No statistically significant differences were observed between ChatGPT-4o and Gemini 2.0 across any of the evaluated metrics (all *p* > 0.05) (Table 5).

Spearman’s correlation analyses for each model are presented in Table 6. A significant negative correlation was observed between FKGL and FKRE across all models: r = −0.730 (*p* = 0.002) for ChatGPT-3.5, r = −0.764 (*p* = 0.001) for ChatGPT-4o, and r = −0.838 (*p* < 0.001) for Gemini 2.0. Additionally, a significant positive correlation between reliability and usefulness was found in all three models: r = 0.870 (*p* < 0.001) for ChatGPT-3.5, r = 0.546 (*p* = 0.035) for ChatGPT-4o, and r = 0.714 (*p* = 0.003) for Gemini 2.0. No other statistically significant correlations were detected among the remaining variable pairs (all *p* > 0.05).

## 4. Discussion

In the current study, the Google Gemini 2.0 model demonstrated a significantly better FKRE score compared to ChatGPT-3.5 and ChatGPT-4o. Additionally, the model achieved significantly better FKGL scores than ChatGPT-3.5 and ChatGPT-4o, indicating enhanced readability. Furthermore, both reliability and usefulness ratings were significantly higher for ChatGPT-4o and Gemini 2.0 than for ChatGPT-3.5. ChatGPT-4o also exhibited greater semantic similarity compared to ChatGPT-3.5. No significant differences in semantic similarity were observed among the other models. A negative correlation was found between the FKRE and FKGL scores across all models. In addition, a positive correlation was observed between reliability and usefulness ratings in all three models, while no significant correlations were identified among the other parameters.

Ankylosing spondylitis is an autoimmune chronic inflammatory disease often seen in young men [24]. Environmental factors such as genetic and epigenetic factors, autoimmune antibodies, abnormal inflammation, stress, and gut dysbiosis are the most frequent causes of this disease. Mechanical stress, vitamin D deficiency, low estradiol levels, infections like Klebsiella pneumoniae, and oxidative stress have also been found to be among the risk factors [25,26,27,28]. In addition to spinal symptoms, the disease may also be accompanied by findings such as peripheral findings, psoriasis, and inflammatory bowel disease [29].

It has been determined that there are delays in diagnosing ankylosing spondylitis over time. This disease is grouped under the umbrella of spondylarthropathies, and the definition is divided into two: Axial spondyloarthropathy and non-radiographic spondyloarthropathy [30,31]. This distinction has contributed greatly to early diagnosis and initiation of treatment.

In addition to these developments, the diagnostic criteria and treatment guidelines have been constantly updated by the EULAR and ASAS [4,5,6]. This has allowed the patient and doctor to make decisions on treatment in light of scientific updates. The disease activity has been controlled easily, deformities have been prevented, and the patients’ quality of life has been improved significantly [3]. In addition to the renewal of diagnostic criteria and treatment guidelines, many new TNF-blocking agents have come into use in pharmacological treatment. Also, more emphasis has been placed on non-pharmacological treatments and rehabilitation methods, resulting in more success in treatment [32,33].

The developments are not limited to this. Artificial intelligence has been started to be used in the field of rheumatology [9]. Possible diagnoses began to be questioned by both patients and healthcare professionals by entering symptoms [10]. The test results were entered, and a diagnosis was requested. Furthermore, radiology images were uploaded to these programs, and these programs were asked to rank the possible diagnoses [10,11,12].

Artificial intelligence programs such as ChatGPT and Gemini were developed by Open AI and Google as large language models. These programs can find the appropriate answer by connecting to various databases when a question is asked of them, and moreover, when feedback is given to these answers, they can memorize it and produce a higher-level answer. This feature has made these programs more and more used day by day. New versions of the programs have also become available [34,35,36]. However, the increasing use of these programs has raised questions about how accurate, safe, and scientific the information they provide is [37]. In light of these questions, studies have been carried out on the use of artificial intelligence in rheumatology by healthcare professionals [38,39].

Nevertheless, model behavior is also influenced by technical parameters such as prompt formulation, temperature settings, and context window size. While these factors were standardized and held constant in our study to ensure methodological uniformity, their potential to impact the structure, accuracy, and creativity of model outputs has been demonstrated in prior research [40]. Future studies may benefit from systematically evaluating how such parameters affect clinical usefulness and information fidelity in medical domains.

Liu et al. examined 36 suggestions generated by ChatGPT and 29 suggestions made by humans for 7 different emergency situations. They underlined that ChatGPT had significant potential in learning from human feedback and giving the right advice in emergency situations [31]. Ren et al. evaluated the ChatGPT-4o and Kimi programs in a cross-sectional study involving 178 patients with ankylosing spondylitis and 4 rheumatologists. They prepared 15 questions related to AS from the literature and analyzed the programs’ answers and revealed these programs as understandable and useful [38]. Chen et al. performed a study on graduate students and checked ChatGPT’s usefulness, novelty, trust, and risk about medical and non-medical issues, and they determined that the program was found to be useful, but there was concern about its usage in clinical practice [37]. Kurshe et al. entered patients’ data into the ChatGPT-4o program, and they found that the preliminary diagnosis predictions of the ChatGPT-4o version were more accurate and cost-effective than the rheumatologists’ [41]. In the current study, the responses of ChatGPT-4o and Google Gemini were significantly more reliable and useful than those of ChatGPT-3.5.

In Coskun et al.’s study, twenty-three questions from an earlier study related to MTX were directed to ChatGPT-3.5 and ChatGPT-4. The accuracy of GPT-4 was found to be 100%, while GPT 3.5’s accuracy was revealed as 86.96%. It was concluded that the ChatGPT-4 program provided more accurate information about methotrexate use [42]. In a cross-sectional study conducted by Efe et al. on ChatGPT-4o and Google Gemini with 420 Board Vitals question bank questions, the accuracy rate of ChatGPT was found to be 86.9%, and that of Gemini was found to be 60.2%. When feedback was given, it was found that these rates increased to 86.7% and 60.5% [43]. Tong et al. asked ChatGPT-3.5, ChatGPT-4o, and Google Gemini 25 open-ended questions derived from the 2022 American College of Rheumatology Guideline prepared for glucocorticoid-induced osteoporosis. Google Gemini was more accurate than the ChatGPT programs, but when feedback was given, the ChatGPT programs were better than Gemini [12]. In the current study, ChatGPT-4o and Google Gemini programs’ answers given to open-ended questions created from the ASAS–EULAR 2022 guideline showed better reliability and usefulness results. In addition to high usefulness and reliability scores, better FKRE and FKGL scores showed that the ChatGPT-4o (*p* = 0.001) and Google Gemini 2.0 programs (*p* = 0.001) were significantly more readable than ChatGPT-3.5. However, in this study, no feedback was given to the programs, and statistical analysis was not performed on new answers. Although post-feedback analysis was not performed in this study, semantic and superficial similarity with ROUGE-L and USE were examined. ChatGPT-4o demonstrated significantly higher semantic similarity scores than ChatGPT-3.5. No significant semantic similarity was observed between the other programs.

Although these programs provide accurate information, educate patients, and suggest treatment options to clinicians, the biggest risk of these programs is undoubtedly the possibility of providing false information [30]. Moreover, inherent limitations of LLMs such as output variability, susceptibility to generating hallucinated or factually incorrect content, dependency on potentially outdated or biased training data, and a general lack of explainability should be carefully considered, especially in clinical contexts [44]. Ethical issues, misuse, responsibility of decisions, patient confidentiality and trust are among other reported risks [33]. During the evaluation, ChatGPT-3.5 failed to mention the ASDAS index alongside ASAS-HI in response to the second question and omitted the BASDAI and ASAS-HI thresholds required in the third question. This oversight contributed to a lower overall score. The primary reason for the significant point deduction was the failure to address the treat-to-target strategy. Furthermore, both ChatGPT-4o and Gemini lost points in the fourth question due to the omission of physiotherapy, a key component of management. Between the sixth and tenth questions, there were several critical errors in ChatGPT-3.5’s responses. Although it mentioned the use of opioids, it neglected to address the associated risk of addiction. It also recommended combining glucocorticoids with other medications, including conventional disease-modifying antirheumatic drugs (DMARDs) such as methotrexate, in the treatment approach and suggested their use in combination with biologic DMARDs, practices that are inconsistent with the current guidelines. Additionally, it recommended IL-23 inhibitors, despite their absence from the approved treatment algorithm. Although the combination of bDMARDs and DMARDs may be preferred in rheumatoid arthritis, such strategies are not supported for this disease, making these recommendations substantial inaccuracies and resulting in major point deductions. In response to the eleventh question, ChatGPT-3.5 did not consider the possibility of misdiagnosis, which also led to a considerable loss of points. Moreover, all three AI programs were found to provide inadequate guidance in the context of comorbid inflammatory bowel disease and uveitis. They incorrectly recommended switching to an IL-17 inhibitor following the failure of a first bDMARD, despite evidence supporting the use of an alternative, established bDMARD as a safer option. Additionally, none of the programs acknowledged the lack of sufficient evidence supporting dose reduction strategies for IL-17 inhibitors and Janus kinase inhibitors.

The strengths of the study are that it compared the latest versions of the programs and objective evaluations were made in terms of reliability and usability, as well as length of answers, understandability, and superficial and semantic similarity. The main limitation of this study is that no feedback was given when the programs responded, and a second analysis was not performed. In addition, this study was designed with only 15 open-ended questions created from the ASAS-EULAR 2022 guidelines. In the future, further studies may be designed with more questions, feedback may be given, LLM–user dialogue scenarios may be designed, and step-by-step diagnosis and symptom clarification may be supplied. Another important limitation is that usefulness and reliability could vary significantly depending on the evaluator’s clinical background. To minimize this issue, we performed this study with two specialists who had experience of more than 10 years in the specialty. Also, it has been known that LLMs like ChatGPT and Gemini are updated frequently; thus, reproducibility may be affected over time and this issue may affect the statistical results. Although the statistical differences were significant, their clinical implications should be interpreted with caution. For example, while higher reliability and readability scores may facilitate patient understanding and support clinical decision-making, the current findings should be validated in real-world clinical settings before drawing definitive conclusions. As has been known, the biggest risk of these programs is that they might give wrong answers. Although wrong answers were detailed qualitatively, new parameters that objectively show the risks are needed. Additionally, it should be kept in mind that LLMs like ChatGPT and Gemini are updated frequently; thus, reproducibility may be affected over time.

## 5. Conclusions

When open-ended questions derived from the EULAR 2022 recommendations were asked to the ChatGPT-3.5, ChatGPT-4o, and Gemini 2.0 programs, it was determined that ChatGPT-4o and Gemini 2.0 gave more reliable, useful, and readable answers. Although there is a risk of obtaining false information, these programs may be useful in patient education and clinician determination of a road map in the future. Future research may explore additional model iterations. Beyond merely adhering to standardized treatment guidelines, personalized therapeutic strategies may be formulated through the integration of patient-specific feedback and patient scenarios. In the longer term, LLMs could also be integrated into clinical decision support systems to assist healthcare professionals in interpreting guidelines and synthesizing complex data, provided that appropriate validation, oversight, and ethical safeguards are implemented. Further studies are needed.

## Figures and Tables

**Table 1 diagnostics-15-01455-t001:** Classification of the ASAS–EULAR 2022 recommendations.

**1. General Management and Monitoring Strategies (goal setting, individualization, disease monitoring, education SAIDs, etc.)**
Q1 Individualization
Q2 Disease follow-up and monitoring frequency
Q3 Targeted therapy
Q4 Patient education and lifestyle recommendations
**2. Pharmacological Treatment (First Step and Advanced Step) (NSAIDs, analgesics, glucocorticoids, conventional, and biological DMARDs)**
Q5 NSAID use and continuous use
Q6 Paracetamol and opioid-like analgesics
Q7 Glucocorticoid injections and systemic use
Q8 csDMARDs (including sulfasalazine)
Q9 High disease activity: TNFi, IL-17i, JAKi
Q10 Selection in case of concomitant uveitis, IBD or psoriasis
**3. Treatment Response and Adaptation Strategies (no response, change in treatment, remission, secondary failure)**
Q11 What should be done if there is no response to treatment?
Q12 First biologic/tsDMARD failure
Q13 Approach in case of permanent remission
**4. Surgery and Accompanying Conditions (hip prosthesis, spinal fracture, deformity, osteotomy, etc.)**
Q14 Hip and spine surgery
Q15 Non-inflammatory conditions such as spinal fracture

**Table 2 diagnostics-15-01455-t002:** Open-ended questions created from the ASAS–EULAR 2022 guidelines.

Q1	What factors should be considered when individualizing the treatment of patients with axial spondyloarthritis?
Q2	How often should disease monitoring of patients with axial spondyloarthritis be, and what should it include?
Q3	How should treatment be guided in patients with axSpA, and which key factors should be considered when setting and acting on a treatment target?
Q4	What lifestyle-related non-pharmacological strategies should be emphasized in the management of axSpA, and what should patient education include to support these?
Q5	Which treatment should be preferred first for managing pain and stiffness in axSpA, and how should the choice between continuous and on-demand use be made?
Q6	Which treatment options can be considered for residual pain in axSpA when first-line therapies have failed, are contraindicated, or not tolerated?
Q7	How should glucocorticoids be used in the management of axSpA, considering both local inflammation and purely axial disease?
Q8	What is the role of csDMARDs in the treatment of axSpA, and how should their use be guided by clinical presentation and current evidence?
Q9	What treatment options should be considered in axSpA patients with persistently high disease activity despite conventional therapy, and how should eligibility and treatment choice be guided?
Q10	How should the presence of extramusculoskeletal manifestations guide biologic treatment choice in patients with axSpA?
Q11	What should be assessed in axSpA patients who show no response to treatment, and why is re-evaluation important before changing therapy?
Q12	What are the recommended treatment options after failure of the first b/tsDMARD in patients with axSpA, and what factors should be considered when switching therapies?
Q13	What treatment approach should be considered for axSpA patients receiving bDMARDs who achieve sustained remission, and what are the key principles of tapering?
Q14	What surgical interventions can be considered in patients with axSpA involving the hip or spine, and what clinical findings and healthcare settings should guide these decisions?
Q15	Which pathologies should be considered and which tests should be performed when a sudden-onset, non-inflammatory spinal pain occurs in a patient with axSpA?

ASAS: Assessment of SpondyloArthritis international Society. EULAR: European League Against Rheumatism. axSpa: Axial spondyloarthropathy. csDMARDs: Conventional disease-modifying antirheumatic drugs. bDMARDs: Biologic disease-modifying antirheumatic drugs.

**Table 3 diagnostics-15-01455-t003:** The 7-point Likert system for evaluation of reliability and usefulness of Chat Generative Pre-trained Transformer, courtesy of Cuma Uz [13].

Reliability Score	Usefulness Score
1 Completely unsafe: none of the information provided can be verified from medical sources or contains inaccurate and incomplete information.	1 Not useful at all: Unintelligible language, contradictory information and missing important information. Not useful for patients.
2 Very unsafe: most of the information cannot be verified from medical sources or is partially correct but contains important incorrect or incomplete information.	2 Very little useful: Partly clear language is used. Some important information is missing or incorrect. Limited possible use for patients.
3 Relatively reliable: the majority of the information provided is verified from medical scientific sources but there is some important incorrect or incomplete information.	3 Relatively useful: Clear language is used. Most important information is mentioned, but some important information incomplete or incorrect. Useful for patients.
4 Reliable: most of the information provided is verified from medical scientific sources but there is some minor inaccurate or incomplete information.	4 Partly useful: Clear language is used. Some important information is missing or incorrect, but most important information is addressed. Somewhat useful for patients.
5 Relatively very reliable: most of the information provided is verified from medical scientific sources and there is very little incorrect or incomplete information.	5 Moderately useful: Clear language is used and most important information is covered, but some important information is still incomplete or incorrect. Useful for patients.
6 Very reliable: most of the information provided is verified from medical scientific sources and there is almost no inaccurate or incomplete information.	6 Very useful: Clear language is used. All important information is mentioned, but some unimportant information or details are also mentioned. Very useful for patients.
7 Absolutely reliable: All of the information provided is verified from medical scientific sources and there is no inaccurate or incomplete information, or missing information.	7 Extremely useful: Clear language is used and all important information is mentioned. Extremely useful to patients, additional information and resources are also provided.

**Table 4 diagnostics-15-01455-t004:** Median and range (Min–Max) values of evaluation metrics for large language models.

	ChatGPT-3.5	ChatGPT-4o	Gemini 2.0
ROUGE-L F1	13.9 (10.7–18.6)	14.4 (10.8–21.2)	13.3 (8.3–17.1)
FKGL	17.8 (14.6–19)	17.1 (15–20.3)	15.7 (13.6–17.9)
FKRE	6.74 (−0.1–20.48)	4.17 (−2.94–16.73)	15.31 (1.13–27.01)
Reliability	4 (3–6)	6 (5–7)	6 (4–7)
Usefulness	5 (4–7)	7 (5–7)	7 (4–7)
Semantic Similarity	66.19 (50.83–76.61)	68.89 (53.64–79.57)	68.85 (57.38–80.7)

ROUGE-L F1: ROUGE-L F1 Score FKGL: Flesch–Kincaid Grade Level FKRE: Flesch–Kincaid Reading Ease Reliability: Reliability Score Usefulness: Usefulness Score Semantic Similarity: USE-based Semantic Similarity Score.

**Table 5 diagnostics-15-01455-t005:** Pairwise comparisons of Flesch–Kincaid Grade Level, Flesch–Kincaid Reading Ease, Reliability Score, Usefulness Score, and USE-based Semantic Similarity of large language models across evaluation metrics (*p*-values).

	1 vs. 2	1 vs. 3	2 vs. 3
**ROUGE-L F1**	0.989 *	**0.025 ***	0.074 *
**FKGL**	0.637 *	**<0.001 ***	**0.003 ***
**FKRE**	0.375 *	**<0.001 ***	**<0.001 ***
**Reliability**	**<0.001 ***	**0.001 ^µ^**	0.486 *
**Usefulness**	**0.002 ^µ^**	**<0.001 ***	0.386 ^µ^
**Semantic Similarity**	**0.038 ***	0.064 ^µ^	0.593 *

ROUGE-L F1: ROUGE-L F1 Score FKGL: Flesch–Kincaid Grade Level FKRE: Flesch–Kincaid Reading Ease Reliability: Reliability Score Usefulness: Usefulness Score Semantic Similarity: USE-based Semantic Similarity Score Model identifiers: 1 = ChatGPT-3.5, 2 = ChatGPT-4o, 3 = Gemini 2.0 ^µ^ Wilcoxon Signed-Rank Test * Paired Samples’ t-test *p* < 0.05 statistically significant values are shown in bold.

**Table 6 diagnostics-15-01455-t006:** Spearman’s correlation coefficients (r) and *p*-values between evaluation metrics for each large language model.

			Semantic Similarity	FKGL	FKRE	Reliability	Usefulness
**ROUGE-L F1 ^µ^**	**ChatGPT-3.5**	r	0.089	−0.210	0.198	−0.089	−0.131
		*p*	0.751	0.452	0.478	0.753	0.641
	**ChatGPT-4o**	r	0.211	−0.004	0.055	0.46	0.078
		*p*	0.450	0.987	0.847	0.085	0.781
	**Gemini 2.0**	r	0.071	−0.047	0.221	−0.261	−0.051
		*p*	0.800	0.869	0.428	0.348	0.858
**Semantic** **Similarity ^µ^**	**ChatGPT-3.5**	r	—	0.068	0.029	0.277	0.406
		*p*		0.810	0.919	0.318	0.133
	**ChatGPT-4o**	r		−0.272	0.164	−0.162	0.095
		*p*		0.327	0.558	0.564	0.737
	**Gemini 2.0**	r		0.03	0.02	0.395	0.328
		*p*		0.914	0.940	0.146	0.232
**FKGL ^µ^**	**ChatGPT-3.5**	r	—	—	−0.730	−0.387	−0.395
		*p*			**0.002 ***	0.154	0.145
	**ChatGPT-4o**	r			−0.764	0.311	−0.12
		*p*			0.001	0.259	0.671
	**Gemini 2.0**	r			−0.838	0.459	−0.111
		*p*			<0.001 *	0.085	0.694
**FKRE ^µ^**	**ChatGPT-3.5**	r	—	—	—	0.505	0.504
		*p*				0.055	0.055
	**ChatGPT-4o**	r				−0.514	−0.256
		*p*				0.05	0.357
	**Gemini 2.0**	r				−0.372	0.15
		*p*				0.172	0.594
**Reliability ^µ^**	**ChatGPT-3.5**	r	—	—	—	—	0.870
		*p*					**0.001 ***
	**ChatGPT-4o**	r					0.546
		*p*					**0.035 ***
	**Gemini 2.0**	r					0.714
		*p*					**0.003 ***

ROUGE-L F1: ROUGE-L F1 Score FKGL: Flesch–Kincaid Grade Level FKRE: Flesch–Kincaid Reading Ease Reliability: Reliability Score Usefulness: Usefulness Score Semantic Similarity: USE-based Semantic Similarity Score ^µ^ Spearman’s rank-order correlation * *p* < 0.05 statistically significant.

## Data Availability

The datasets used and/or analyzed during the current study are available from the corresponding author on reasonable request.

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
