# Peer review of "Evaluation of the Performance of Large Language Models in the Management of Axial Spondyloarthropathy: Analysis of EULAR 2022 Recommendations"

_diagnostics, 2025, doi:10.3390/diagnostics15121455_

Round 1

Reviewer 1 Report

Comments and Suggestions for Authors

In this paper titled “Evaluation of the Performance of Large Language Models in the Management of Axial Spondyloarthropathy: Analysis of EULAR 2022 Recommendations”,

the authors have carried out comparison study of three large language models (ChatGPT-3.5, GPT-4o, and Gemini 2.0 Flash) for the responses to open ended questions created from ASAS–EULAR 2022 guidance.  ASAS–EULAR 2022  are recommendations for the management of axial spondyloarthritis (axSpA). The responses were evaluated for their usability,  reliability, Flesch-Kincaid Reading Ease (FKRE) and Flesch-Kincaid Grade Level (FKGL) metrics for readability, Universal Sentence Encoder (USE) and ROUGE-L for semantic and surface-level similarity. Better FKRE and FKGL scores were obtained in the Google Gemini 2.0 program. Reliability and usefulness scores were significantly higher for ChatGPT-4o and Gemini 2.0. During the evaluation the authors have observed that  ChatGPT-3.5 failed to mention the ASDAS index  alongside ASAS-HI in response to the second question and omitted the BASDAI and ASAS288 HI thresholds required in the third question. Both ChatGPT-4o and Gemini lost points in the fourth question  due to the omission of physiotherapy, a key component of management. Between the sixth and tenth questions, several critical errors were noted in ChatGPT-3.5's responses. All the three models were found to provide inadequate guidance in the context of comorbid inflammatory bowel disease and uveitis.

The following questions/corrections needs to be answered/carried out

  1. Under the “Introduction” section, please specify the Objective of study as two or three pointers.
  2. Under the section “Large language Models Used and Response Collection Process”, when the same question was asked to models twice or thrice, was their any difference in responses generated by the models. Discuss on these observations.
  3. Under the section “Evaluation Process of Response”,
  4. Line number 148, for Universal Sentence Encoder (USE), why the text has been encoded by 512-dimensional vectors ?
  5. Provide Equations for the metrics FKRE, FKGL, ROUGE-L
  6. Under the section “Statistical Analysis”, line number 173, elaborate on why p-value of <0.05 was considered significant for this study.
  7. Under the section “Conclusion”, discus with few more details on what “Further studied are needed”.

Author Response

Thank you for your time for considering the study. I tried to complete all of your suggestions and sent you revised form. Thank you for your contribution to the study.

1-The objective of study as two or three pointers were added in introduction. (line 111-113 )

2-In discussion Tong at al.’s study was discussed which used feedback method in their study. (line 288-292 and 296-297) Also the limitation of not giving feedback to the models was discussed in limitations. (line 333-338)

3,4- Thank you for your valuable comment. The 512-dimensional output is a standard and intentional design choice of the Transformer-based Universal Sentence Encoder. This fixed dimensionality was empirically selected to provide an optimal balance between semantic richness and computational efficiency. While higher dimensions such as 1024 could theoretically represent more information, they require significantly more memory and processing time, often without meaningful performance gain in practical NLP tasks. The 512-dimensional vectors enable robust semantic similarity analysis while remaining computationally lightweight and widely applicable. We have added a clarifying sentence to the manuscript and cited the original publication accordingly. (line 158, 159, reference 14)

5- Thank you for your suggestion. We have added the explicit mathematical formulas for the FKRE, FKGL, and ROUGE-L F1-score to the relevant section of the manuscript. These reflect the standard definitions used in readability and surface similarity analyses. The FKRE and FKGL metrics were calculated using the textstat Python library, while the ROUGE-L scores were computed using the rouge-score package. These additions enhance the transparency and reproducibility of our evaluation methodology. The formulas used are as follows:

“FKRE = 206.835 − 1.015 × (total words / total sentences) − 84.6 × (total syllables / total words) FKGL = 0.39 × (total words / total sentences) + 11.8 × (total syllables / total words) − 15.59 ROUGE-L F1 = [(1 + β²) × P × R] / [R + β² × P],where P = LCS / m, R = LCS / n, and β = P / R”  (line 175-178)

6- Thank you for your comment. We have added a brief explanation to clarify the rationale for using a p-value threshold of  < 0.05. This value is a conventional standard in biomedical research, used to control the probability of a Type I error (false positive) while balancing statistical sensitivity. This clarification has been added to the end of the “Statistical Analysis” section of the manuscript. (line 189-191)

7- Details of further studies were added to conclusion sectioion. (line 357-363)

Thank you for all of the suggesions. If you have any more suggesions do not hesitate to contact me. I am looking forward to your final decision and publishing the study in your journal.

Yours sincerely.

Reviewer 2 Report

Comments and Suggestions for Authors

The study titled “Evaluation of the Performance of Large Language Models in the Management of Axial Spondyloarthropathy: Analysis of EULAR 2022 Recommendations” investigates the effectiveness of large language models (LLMs) such as ChatGPT-3.5, ChatGPT-4o, and Google Gemini 2.0 in responding to open-ended clinical questions derived from the ASAS–EULAR 2022 guidelines for axial spondyloarthropathy (axSpA) management. A total of fifteen open-ended questions, created from key guideline recommendations, were posed to each model. The responses were evaluated using both qualitative and quantitative metrics, including readability (Flesch-Kincaid Reading Ease and Grade Level), semantic and surface-level similarity (Universal Sentence Encoder and ROUGE-L), and clinical reliability and usefulness (assessed via a 7-point Likert scale by medical professionals). The results showed that Google Gemini 2.0 produced the most readable responses, while both ChatGPT-4o and Gemini 2.0 were rated significantly higher in terms of reliability and usefulness compared to ChatGPT-3.5. ChatGPT-4o also demonstrated superior semantic alignment with the guideline content. The study concludes that ChatGPT-4o and Gemini 2.0 hold potential to support clinicians and patients in making informed decisions in rheumatology, although caution must be exercised due to the possibility of incorrect or incomplete outputs. Further research is recommended to validate and enhance the clinical application of LLMs in practice.

The topic is timely and significant as LLMs are increasingly being explored for clinical decision support. Assessing LLM responses against validated clinical guidelines (EULAR 2022) ensures contextual relevance. However, while the concept is relevant, similar LLM evaluations are already emerging in the literature. The study adds specificity by focusing on axial SpA. Therefore, the topic is justified, subject to the following improvements.

  1. Minor grammar and formatting polishing, especially in the discussion and conclusion.
  2. Only 15 questions were analysed, which may not be sufficiently representative of the full guideline's complexity.
  3. Ratings on usefulness and reliability could vary significantly depending on the evaluator’s clinical background.
  4. There is no discussion on how model prompts, temperature settings, or context windows might affect the responses.
  5. LLMs like ChatGPT and Gemini are updated frequently; thus, reproducibility may be affected over time.
  6. The statistical significance of results (p < 0.05) is reported but not deeply discussed in terms of clinical significance.
  7. Correlations between readability and perceived reliability/usability are insightful, suggesting that clinicians may prefer clearer, simpler text.
  • My suggestion is to incorporate a wider range of clinical scenarios and more evaluators (including patients). Also, evaluate real-world performance in clinical simulations or EMR-integrated settings, and provide transparency in how prompts were phrased and whether any prompt engineering was applied.

Author Response

Thank you for your time for considering the study. I tried to complete all of your suggestions and sent you revised form. Thank you for your contribution to the study.

1-Minor grammer mistakes were corrected. (line 53, 54, 101, 216-221, 229-234, 240,241, 243-246, 247-251, 256-260, 276-280, 314, 315, 321-323, 331, 332, 347, 348). Some sentences were written more clearly. Thank you for your contribution to the study.

2-In this study, all recommendations in the 2022 ASAS–EULAR guideline for the management of axial spondyloarthritis (a total of 15 recommendations) were directly transformed into open-ended clinical questions based on the reference text. Each question was evaluated both individually by experienced physiatrists and objectively through software-based metrics such as ROUGE, USE, FKRE, and FKGL. Therefore, our study does not represent only a portion of the guideline, but rather the entire content, and it reflects the comprehensive structure of the guideline both methodologically and in terms of content. Of course, it would have been possible to generate more than one question per recommendation, resulting in a total of 30 or more questions. However, this would have required fragmenting the clinical message of a single recommendation and distributing it across multiple questions, which could compromise the integrity of the reference content and potentially jeopardize the objectivity of the evaluation process.

In contrast, in our study, each question was carefully constructed to directly reflect the core clinical intent of the corresponding recommendation, thereby preserving both methodological consistency and content integrity.

In conclusion, the fact that the number of analyzed questions is limited to the number of recommendations has not reduced the representativeness of the study. On the contrary, it has enabled a systematic, balanced, and comprehensive evaluation of all the recommendations in the ASAS–EULAR guideline.

3-It is true that ratings on usefulness and reliability could vary significantly depending on the evaluator’s clinical background. Because of this issue we perfomed the study with 2 specialists who had experience more than 10 years in speciality and examining rheumatic patients everyday. Thank you for underlying this point. We also underlined the issue in limitations. (line 338-341) Thank you for your contribution to the study.

4-Thank you for this insightful comment. We agree that technical parameters such as prompt structure, temperature settings, and context window size may have a considerable impact on the outputs generated by large language models (LLMs). Although these variables were standardized across all models in our methodology to ensure a uniform comparison, we acknowledge the importance of discussing their potential influence in the manuscript. Therefore, we have added a new paragraph as the 6th paragraph of the discussion section, addressing this point and citing relevant literature. (line  261-266,  reference 41)

5-It is true that LLMs like ChatGPT and Gemini are updated frequently; thus, reproducibility may be affected over time. Thank you for underlying this point. We also underlined the issue in limitations. (line 341-344) Thank you for your contribution to the study. To mitigate this issue, each question was presented to the models only once. The responses were printed in duplicate and distributed to the specialists, who evaluated the models independently, without knowledge of which response corresponded to which model. This issue was undelined in methods section also. (line 151-155) Thank you for your contribution to the study.

6-Thank you for this thoughtful comment. We agree that statistical significance does not always imply clinical significance. To address this point, we have added a brief statement at the end of the paragraph discussing the study’s limitations in the Discussion section. This addition highlights that the clinical implications of statistically significant differences such as improvements in reliability, usability, and readability should be interpreted with caution. We also emphasized the need for validation in real-world clinical settings. (line 343-347)

7-It is true that LLMs like ChatGPT and Gemini are updated frequently; thus, reproducibility may be affected over time. Thank you for underlying this point. We also underlined the issue in limitations (line 341-343) Thank you for your contribution to the study.

Thank you for all of the suggesions. If you have any more suggesions do not hesitate to contact me. I am looking forward to your final decision and publishing the study in your journal.

Yours sincerely.

Reviewer 3 Report

Comments and Suggestions for Authors

Comments for authors are in the document.

Author Response

Thank you for your time and considering the study.  

If you have any more suggesions do not hesitate to contact me. I am looking forward to your final decision and publishing the study in your journal.

Yours sincerely.

Reviewer 4 Report

Comments and Suggestions for Authors The aim of this article is to evaluate the capabilities of large language models (LLMs) such as ChatGPT-3.5, ChatGPT-4o, and Gemini 2.0 in interpreting and justifying clinical guidelines for the treatment of axial spondyloarthritis (axSpA) based on ASAS–EULAR 2022 documents. To achieve this, the authors developed a methodology that included transforming official recommendations into open-ended questions, generating responses using LLMs, and performing both expert and automated evaluations across multiple parameters, which enabled them to draw sufficiently convincing conclusions. The topic of this study is relevant and aligns with current trends in the diagnosis of complex diseases. Its results carry a high level of practical significance. The combination of expert and algorithmic assessments, as well as the detailed analysis of LLM errors, indicates a comprehensive methodological approach. At the same time, it should be noted that the cross-sectional study design selected by the authors is not fully suitable for a comprehensive analysis of the research problem, and the obtained results should be considered only as an initial starting point for further studies. Additionally, the significance of the study is diminished by the absence of a feedback stage for the LLMs and a subsequent regeneration of responses, as this contradicts one of the key principles of modern LLMs — reinforcement learning and reflexive improvement. Another limitation is the small sample size of only 15 questions, which restricts statistical power and the generalizability of the results. The use of models in independent sessions represents a suboptimal dialogue scenario without context or follow-up questions and does not simulate real clinical practice. Thus, the chosen study design substantially limited the potential capabilities of LLMs, which are best revealed in diverse prompts and deeper user-model interaction. The article presents a sufficiently comprehensive and up-to-date literature review (including publications from 2023 to 2025), covering both works on the effectiveness of LLMs in medicine and studies related to axSpA and EULAR guidelines. However, it remains unclear why the authors presented the literature review in the Discussion section rather than in the Introduction, as is standard in the structure of classical scientific papers. This approach makes it impossible to formulate a research aim justified by the current state of the problem. It should also be noted that the process of interpreting the study results is difficult to understand, which reduces the article’s readability and significantly limits the number of potentially interested readers. The terminology used by the authors is accessible only to specialists in the field of statistical analysis. Nevertheless, it can be acknowledged that the article presents the results of a completed study and can be recommended for publication after certain revisions. Authors are advised to: Move the literature review and the formulation of the study aim to the Introduction section. Expand the Discussion section to describe the limitations of LLM generation, including issues of output instability, the possibility of hallucinations, limitations of training databases, and lack of explainability. Describe possible LLM-user dialogue scenarios that are closer to real clinical practice (e.g., step-by-step dialogues, symptom clarification). In the Conclusion section, consider the prospects for using LLMs in clinical decision support systems. Briefly discuss legal and ethical aspects (risk of misinformation, liability for recommendations, patient trust). Provide a tentative roadmap for future research, including, for example, a larger number of questions, inclusion of real clinical cases, regeneration of responses after feedback, evaluation of LLM output reproducibility, and more.

Author Response

 Thank you for your time for considering the study. I tried to complete all of your suggestions and sent you revised form. Thank you for your contribution to the study.

  1-The literature review and the formulation of the study aim was moved to the Introduction section. (line 98-104, references 9-12) Thank you for your contribution to the study.

    2-Thank you for this insightful and constructive comment. We agree that addressing the fundamental limitations of large language models (LLMs) is crucial, especially in clinical research and applications. In response, we have expanded the Discussion section by incorporating a sentence that acknowledges these key limitations. Specifically, we added the following sentence to the seventh paragraph of the Discussion section, immediately after: “However, in this study, as in other studies, feedback was not given to the programs, and statistical analysis was not performed on new answers.” (line 296-297) The inserted sentence reads: “Moreover, inherent limitations of LLMs such as output variability, susceptibility to generating hallucinated or factually incorrect content, dependency on potentially outdated or biased training data, and a general lack of explainability should be carefully considered, especially in clinical contexts (Azamfirei et al., 2023).” This addition helps contextualize the study findings by highlighting broader concerns about the reliability and safety of LLM-generated content in clinical decision-making. (line 302-308, reference 45)

3-LLM-user dialogue scenarios designed to mimic real-world clinical practice, particularly in the context of step-by-step diagnosis and symptom clarification were discussed in discussion. (line 336-338) Thank you for your contribution to the study.

4-Thank you for this thoughtful comment. We agree that large language models (LLMs) have potential to contribute meaningfully to clinical decision support systems (CDSS). In response, we have expanded the conclusion section to briefly discuss this potential. A sentence has been added to conclusion the end of the paragraph to emphasize how LLMs could serve as complementary tools for clinicians, while also noting the importance of further validation and safeguards in clinical environments. (line 357-363)

5-  In conclusion the prospects for using LLMs in clinical decision support systems were added. (360-362) Thank you for your contribution to the study.

6- Legal and ethical aspects (risk of misinformation, liability for recommendations, patient trust…) were discussed. (line 302-308) Thank you for your contribution to the study.

7-A tentative roadmap for future research, including, for example, a larger number of questions, inclusion of real clinical cases, regeneration of responses after feedback, evaluation of LLM output reproducibility was given in coclusion. (line 357-363) Thank you for your contribution to the study.

Thank you for all of the suggesions. If you have any more suggesions do not hesitate to contact me. I am looking forward to your final decision and publishing the study in your journal.

Yours sincerely.
